# Monitoring Visual Fatigue with Eye Tracking in a Pharmaceutical Packing Area

**DOI:** 10.3390/s25185702

**Published:** 2025-09-12

**Authors:** Carlos Albarrán Morillo, John F. Suárez-Pérez, Micaela Demichela, Mónica Andrea Camargo Salinas, Nasli Yuceti Miranda Arandia

**Affiliations:** 1Applied Science and Technology Department, Politecnico di Torino, 10129 Turin, Italy; micaela.demichela@polito.it; 2Escuela de Ingenieria y Ciencias, Tecnologico de Monterrey, Ave. General Ramon Corona 2514, Zapopan 45138, Mexico; jf.suarez@tec.mx; 3Departamento de Ingeniería Industrial, Universidad de América, Bogotá 111321, Colombia; monica.camargo@uamerica.edu.co (M.A.C.S.); nasli.miranda@uamerica.edu.co (N.Y.M.A.)

**Keywords:** visual fatigue, eye tracking, wearable sensors, human-centered design, pharmaceutical packaging, Industry 5.0, occupational health

## Abstract

**Highlights:**

**What are the main findings?**
This study introduces a robust methodology for assessing visual fatigue in real industrial environments, specifically in a pharmaceutical packaging area by integrating subjective reports, biometric eye-tracking data, and environmental lighting measurements.Results reveal a consistent increase in visual fatigue across the workweek and shifts, with critical fatigue peaks during night shifts and tasks involving high physical and visual demand under suboptimal lighting.

**What is the implication of the main finding?**
Combining physiological, perceptual, and environmental data addresses a major gap in real-time visual fatigue monitoring on the shop floor, moving beyond control room and laboratory-based assessments.The approach contributes to Industry 5.0 goals by enabling adaptive, human-centered interventions that enhance operator well-being, task accuracy, and sustainable ergonomic design.

**Abstract:**

This study investigates visual fatigue in a real-world pharmaceutical packaging environment, where operators perform repetitive inspection and packing tasks under frequently suboptimal lighting conditions. A human-centered methodology was adopted, combining adapted self-report questionnaires, high-frequency eye-tracking data collected with Tobii Pro Glasses 3, and lux-level measurements. Key eye-movement metrics—including fixation duration, visit patterns, and pupil diameter—were analyzed within defined work zones (Areas of Interest). To reduce data complexity and uncover latent patterns of visual behavior, Principal Component Analysis was applied. Results revealed a progressive increase in visual fatigue across the workweek and throughout shifts, particularly during night work, and showed a strong association with inadequate lighting. Tasks involving high physical workload under poor illumination emerged as critical risk scenarios. This integrated approach not only confirmed the presence of visual fatigue but also identified high-risk conditions in the workflow, enabling targeted ergonomic interventions. The findings provide a practical framework for improving operator well-being and inspection performance through sensor-based monitoring and environment-specific design enhancements, in alignment with the goals of Industry 5.0.

## 1. Introduction

Visual fatigue, also referred to as eye strain or visual discomfort, is a physical and cognitive condition that arises from prolonged visual activity, particularly under demanding or non-ergonomic conditions [1]. It commonly occurs when individuals focus on the same target for extended periods, engage in frequent eye movements, or process large volumes of visual information [2]. Symptoms include ocular discomfort, dryness, blurred vision, headaches, and diminished concentration, all of which can negatively affect cognitive functioning and overall task performance [3,4]. These impacts are increasingly relevant as industries transition toward more intelligent and human-centric paradigms, such as those promoted by Industry 5.0, which emphasizes worker well-being and adaptive technology [5].

This type of fatigue presents a significant challenge in occupations that require sustained visual precision and attention to detail. It slows performance, increases the likelihood of missing critical signals, and elevates the risk of errors. Such challenges are particularly problematic in sectors such as healthcare, manufacturing, packaging, and transportation. For example, a quality control operator scanning hundreds of products or a technician inspecting small defects under bright lighting may experience reduced accuracy when visual fatigue sets in [6]. Despite its consequences, visual fatigue is often underestimated or accepted as inevitable. However, its cumulative effects can lead to chronic stress, long-term eye discomfort, and early manifestations of computer vision syndrome (CVS) [7,8]. This not only compromises worker comfort but also reduces performance and jeopardizes health—an issue that industries can no longer afford to ignore.

The urgency of this problem becomes clearer when examined in the context of regional and global data. According to the Colombian Safety Council (2024) [9], the manufacturing sector accounted for the highest incidence of occupational diseases in 2024 (21.1% of cases), followed by administrative and support service activities (12.4%). Musculoskeletal disorders remain the most frequently reported category [10]. Although visual health problems are increasingly recognized, the Colombian legal framework (Decree 676 of 2020) does not explicitly classify CVS or visual fatigue as occupational diseases, instead emphasizing more traditional ocular conditions such as conjunctivitis, keratitis, and cataracts. Nevertheless, the legislation allows these emerging conditions to be recognized if a clear cause–effect relationship is demonstrated. International evidence also points to substantial productivity losses: one study in Saudi Arabia [11] found that 7.6% of workers reported absences due to fatigue-related dry eye, while 29.4% indicated that it reduced their effective working hours. Globally, the World Health Organization (WHO) and the International Labour Organization (ILO) estimate that uncorrected vision problems result in an annual productivity loss of approximately USD 411 billion [12]. Within Colombia, a study of pharmaceutical workers in Bogotá reported a 51.4% prevalence of CVS [13], underscoring the scale of the problem in industrial contexts. High prevalence rates have also been documented among academic populations: a 2022 study of medical students in Tunja reported an 84.4% prevalence of CVS [14], while research with optometry students at La Salle University in Bogotá found headache (89.9%), red eyes, and dry eyes as the most frequent symptoms [15]. Despite these findings, the true impact of visual fatigue often remains obscured in absenteeism records, where it is misclassified as musculoskeletal disorders or general illness.

This context highlights a critical gap in both research and workplace practice, which the present study seeks to address. Despite increasing awareness, visual fatigue continues to be underreported and insufficiently measured, particularly in operational environments. A comprehensive understanding requires robust, multidimensional assessment strategies. However, this is not straightforward: fatigue manifests differently among individuals and does not always produce obvious or immediate signs [4]. Some workers may perceive eye heaviness, blurred vision, or difficulty maintaining concentration, while others may remain unaware, continuing their tasks as performance silently deteriorates. Often, by the time fatigue is consciously recognized, significant cognitive and perceptual decline has already occurred. For this reason, researchers must adopt integrative approaches that combine subjective perceptions with physiological data using smart technologies, in line with Industry 5.0’s emphasis on responsive, worker-centered systems.

Subjective assessments provide valuable insights into the personal experience of visual fatigue. Tools such as the Visual Fatigue Scale (VFS) [16] or the Simulator Sickness Questionnaire (SSQ) [17] ask straightforward questions—for example: Are your eyes sore or tired? Are you struggling to focus? Do you feel mentally drained or slightly dizzy? These questionnaires are quick and easy to administer, offering useful information on how individuals cope with task demands. However, responses can be inconsistent: some participants underreport their symptoms, while others overreport, and subtle signs of fatigue often remain undetected.

Objective tools complement these limitations by capturing underlying physiological changes. Eye trackers and related devices provide rich data on visual behavior, including fixation duration, saccadic movements, blinking frequency, and pupil responses to light and stress [18]. These subtle indicators often reveal the early manifestations of visual fatigue before the individual becomes consciously aware of them [19]. More advanced approaches incorporate electroencephalography (EEG) to directly monitor brain activity [20]. By tracking changes in brainwave patterns, EEG can detect increases in mental workload or lapses in attention. For example, higher proportions of slower theta waves or reductions in faster alpha waves are indicative of mental fatigue and reduced attentional capacity [21]. Although EEG is not yet practical for widespread workplace use, it remains a valuable laboratory tool, particularly when combined with eye tracking or heart rate monitoring [22]. Integrating these multimodal insights advances the understanding of visual fatigue and enables the development of early-intervention strategies.

In addition to these physiological measures, conventional approaches such as critical flicker fusion frequency (CFF) and accommodative function tests have been widely used as indicators of visual fatigue. While these methods are effective in controlled laboratory or clinical settings, they present practical limitations for industrial environments: they require specialized equipment, interrupt workflow, and are not suitable for continuous or non-intrusive monitoring. By contrast, eye-tracking measures can be integrated seamlessly into real work settings, allowing fatigue to be assessed dynamically without disrupting production tasks. This distinction underscores the value of the present study, which emphasizes real-time, context-sensitive monitoring aligned with the human-centric principles of Industry 5.0.

Until now, much of the knowledge on visual fatigue has been derived from domains where sustained attention is critical. Research has expanded in areas such as gaming, professional driving—including truck drivers, train operators, and pilots—and immersive technologies such as virtual and mixed reality (VR/MR). In these contexts, performance depends heavily on the ability to maintain prolonged visual engagement without loss of accuracy or speed [23]. Other lines of research have focused on 3D technologies, particularly in training and simulation, where depth perception and screen-based interaction introduce new visual demands [24]. Office work has also been a major concern, with prolonged exposure to screens being closely linked to digital eye strain, now recognized as a major occupational health issue [25]. Long-term use of visual display terminals (VDTs) has been associated with discomfort, reduced productivity, and risks to long-term eye health [26]. Studies in healthcare environments highlight similar issues for diagnostic and monitoring tasks that rely on digital interfaces, while cultural and museum settings are beginning to examine how interactive, screen-based exhibits influence visitor fatigue and engagement [27].

Despite these advances, studies specifically targeting industrial environments remain limited. Our review identified 21 relevant articles [28,29,30,31,32,33,34,35,36,37,38,39,40,41,42,43,44,45,46,47,48]. Most of these focus on control rooms, monitoring stations, or screen-intensive roles, where operators engage in prolonged observation of digital displays. These include process industry control rooms, refineries, and display-dominated workstations, with several studies addressing the influence of lighting and shift schedules on visual strain [33,34,36,37,41]. Other contributions have explored lighting ergonomics, visual comfort, or physiological responses such as EEG and pupil dynamics, though often within laboratory settings disconnected from actual industrial workflows [32,38,40,42,44,48]. A small number of exploratory studies have tested augmented reality systems, smart wearables, or fatigue detection algorithms in industrial applications, but these remain experimental and not yet fully embedded in operational practice [29,31,42,43].

Only a few studies have directly examined hands-on, visually demanding activities on the shop floor, beyond control room settings. These tasks are typically characterized by suboptimal lighting, strict time pressures, and high repetition [28,30,35,39,45,46,47]. Among them, only a handful have integrated smart technologies—such as eye tracking, fatigue prediction algorithms, or sensor-driven lighting systems—to objectively monitor visual fatigue in real time [35,45,46,47].

This highlights a critical research gap: very few studies have investigated visual fatigue in operational industrial environments using real-time, objective, and accurate measurements. Smart technologies are particularly valuable because they enable continuous, non-intrusive data collection and provide early warning of fatigue, often before observable declines in performance occur. These tools are essential for advancing Industry 5.0, which emphasizes the augmentation of human capabilities through adaptive technologies that support the worker, rather than requiring the worker to adapt to the system [49]. They are central to human-centered design, early intervention, and proactive strategies that enhance both operator well-being and process reliability [50].

In response to this gap, the present study investigates visual fatigue in a pharmaceutical packaging line, applying a human-centered, sensor-based methodology. By combining environmental assessments, operator feedback, and objective eye-tracking data collected with Tobii Pro Glasses 3, the study captures how visual fatigue develops during real work shifts. This approach enables the identification of high-risk tasks, a deeper understanding of fatigue dynamics in context, and the formulation of evidence-based ergonomic interventions to improve operator well-being and inspection performance.

The following sections describe the methodology adopted in this study, outlining the experimental design and procedures used to assess visual fatigue in a real industrial environment. This is followed by the presentation and discussion of results, focusing on the identification of critical risk points in the workflow. The paper concludes with a summary of key findings and their implications for human-centered industrial design and fatigue prevention.

## 2. Materials and Methods

This section outlines the methodology adopted to investigate visual fatigue in a real-world pharmaceutical packaging environment. Rather than relying on controlled simulations or laboratory experiments, the study was designed to capture how visual fatigue develops naturally during regular work shifts. The following subsections describe the experimental setup, data collection procedures, and analytical methods used to identify fatigue patterns and highlight critical points in the packaging process where ergonomic interventions could have the greatest impact.

### 2.1. Experimental Setup

The study was carried out in the packaging area of a pharmaceutical company, where workers routinely perform tasks requiring sustained visual attention, such as checking, packaging, and sealing pharmaceutical products. These tasks involve grouping items into units of one, ten, or one hundred or on large trays and demand continuous visual focus, rapid hand movements, and precise motor control. This environment provided an ideal context for investigating visual fatigue under realistic working conditions. To illustrate the setup, Figure 1 presents a representative heatmap generated from one participant’s eye-tracking data. The visualization shows how visual attention was concentrated on the trays of pharmaceutical products prior to packaging (areas in read), under real shop-floor conditions.

To capture both objective and subjective indicators of visual fatigue, we employed the Tobii Pro Glasses 3 eye-tracking system in combination with a modified Visual Fatigue Scale questionnaire. The Tobii Pro Glasses 3 (Tobii AB, Danderyd, Sweden) is a lightweight, wearable device capable of sampling at up to 100 Hz, recording detailed eye-movement data—including gaze behavior, fixations, saccades, and pupil diameter—within natural industrial environments. Specific Areas of Interest (AoIs) were defined within the workspace, focusing on stations dedicated to the inspection and packaging of vials and packs. The eye-tracking data collected within these AoIs included the following parameters:Total Fixation Duration: Cumulative time fixating on the AoI.Fixation Count: Total number of fixations.Total Visit Duration: Total time spent in the AoI per visit.Average Visit Duration: Mean time per visit within the AoI.Visit Count: Number of gaze entries into the AoI.Percentage of Gazes as Fixations: Share of gazes resulting in fixations.Percentage of Total Activity Time in the AoI.Pupil Diameter: Monitored during fixations to infer cognitive load and fatigue.

Subjective visual fatigue was measured using an adapted Visual Fatigue Scale [16], comprising three Likert-scale items:Eye tirednessVision clarityEye discomfort

Each item was rated on a scale from 0 (very fresh/clear) to 4 (very tired/blurred), yielding a total score ranging from 0 to 12.

Because the success of visual inspection largely depends on the operator’s eyesight, the illumination system is critical for ensuring optimal performance. To contextualize the eye-tracking data, environmental lighting levels (measured in lux) were recorded using a luxmeter installed near the AoIs. Measurements were taken once at the beginning of each trial, following prior verification that lighting remained stable throughout the session.

The instruments used to assess visual fatigue included the eye tracker (Tobii Pro Glasses 3), the luxmeter (for illumination levels), and the adapted Visual Fatigue Scale. Together, these tools provided a comprehensive perspective on visual fatigue by integrating subjective perceptions with objective physiological indicators under different operational conditions. Although additional instruments were employed in the broader project to evaluate other human factors—such as stress, general fatigue, and biomechanical risks—these aspects fall outside the scope of this paper.

### 2.2. Data Collection

Before the study commenced, all 43 participants (42 females and 1 male) provided informed consent. The research was approved by the Ethics Committee of Universidad de América (Protocol No. 002-2024), in accordance with the ethical principles of the Declaration of Helsinki. Participation was entirely voluntary, and no financial compensation was offered. With 43 participants out of approximately 80 total operators, the sample represented more than half of the workforce assigned to these packaging tasks. The group was evenly distributed across the four production lines, ensuring balanced representation of different operational settings. The gender distribution also reflected the real workplace context, where only three to four of the ~80 operators are male. Participant ages ranged from 19 to 53 years (mean = 32.2), making this a contextually appropriate and representative sample.

To capture the progression of visual fatigue during a typical shift, each operator was assessed at two key points: once during the first two hours of the shift (before the scheduled break) and once during the final two hours (after the break). These intervals were chosen to respect operational constraints, minimizing disruption to production while allowing meaningful observation of fatigue dynamics. Data were collected continuously throughout the workweek (Monday to Friday) across all three standard 8-h shifts: morning (06:30–14:30), afternoon (14:30–22:30), and night (22:30–06:30). The study was carried out in four production scenarios: Plant 4 Line 2, Plant 4 Line 3, Plant 4 Line 4, and Plant 8. Workers performed their usual inspection, packaging, and sealing activities under real shop-floor conditions, involving individual units, batches of 10 or 100, and large trays. This repetitive and visually demanding workload, combined with differences in lighting and layout among the lines, provided an important context for evaluating environmental influences on visual fatigue.

Each data collection session lasted 20 min, during which participants wore the Tobii Pro Glasses 3 to record detailed eye-movement patterns under natural working conditions. At the beginning of each trial, illumination levels were measured using the luxmeter installed near the AoIs. To complement these objective measures, participants also completed the adapted Visual Fatigue Scale. The questionnaire was administered immediately prior to each eye-tracking session, meaning that every operator provided two subjective self-reports: one during the first assessment period (early in the shift) and one during the second assessment period (toward the end of the shift). Each assessment took only a few minutes and was conducted under the same shop-floor conditions, ensuring minimal disruption to workflow. This procedure ensured that the subjective perception of fatigue was captured in real time and paired directly with objective biometric recordings from eye-tracking and environmental measurements. The dual assessment design (early and late shift) enabled the study to track both the accumulation of fatigue across the day and the weekly trend across shifts, while maintaining consistency and comparability across participants.

### 2.3. Data Analysis Techniques

This section describes the analytical techniques applied to examine the collected data, with the aim of understanding how visual fatigue evolves over the course of a work shift. The analysis integrated environmental factors, biometric indicators obtained from eye-tracking, and responses from the subjective Visual Fatigue Scale to identify the most visually demanding moments in the workflow—those posing the greatest risk to operator comfort, accuracy, and well-being. These insights formed the basis for proposing targeted ergonomic improvements designed to mitigate visual strain and enhance the long-term sustainability of packaging tasks.

#### 2.3.1. Lighting Levels Analysis

Before proposing ergonomic interventions, it was essential to evaluate whether the lighting conditions at each workstation complied with internationally recognized standards for visual tasks. Reference was made to the international lighting standard [51] for indoor work environments, which is also applicable in Colombia. This standard recommends a minimum of 500 lux for precision tasks such as inspection and quality control, while tasks with higher visual demands may require up to 1500 lux to prevent eye strain and ensure sustained performance. Maintaining illumination within this range is critical for minimizing visual fatigue and supporting consistent task execution.

In this study, lighting levels were measured at the beginning of each trial and compared against the ISO/CIE 8995-1:2025 recommended thresholds [51]. Deviations from these values were then analyzed in relation to their potential influence on visual fatigue, allowing the identification of environmental risk factors that could compromise operator comfort and visual performance.

#### 2.3.2. Eye-Tracker Analysis

Using the Tobii Pro Glasses 3, we collected seven key eye-tracking metrics to assess visual behavior, along with pupil diameter as an additional physiological indicator. The metrics included: Total Fixation Duration (Tot Fix Dur), Fixation Count, Total Visit Duration (Tot Visit Dur), Average Visit Duration (Avr Visit Dur), Visit Count, Percentage of Total Activity Time in the AoI, and Percentage of Gazes as Fixations. Each metric was recorded within defined AoIs to capture fluctuations in visual attention and engagement during actual work tasks.

Before incorporating pupil diameter into subsequent analyses, its reliability under the study conditions was evaluated. Because pupil size is highly sensitive to external factors—particularly illumination, but also temperature and humidity [52]—it was necessary to determine whether it could function as a valid indicator of fatigue in this real-world setting. To this end, correlations were examined between pupil diameter and environmental variables, including light intensity (Lux), temperature (T), and relative humidity (RH). This step controlled for potential confounding influences and established whether pupil data could be meaningfully interpreted as a marker of visual strain.

To analyze the relationships among the eye-tracking variables and reduce redundancy, Principal Component Analysis (PCA) was applied. PCA was selected not only to reduce dimensionality but also to address multicollinearity and improve the interpretability of fatigue-related patterns [53]. Preliminary analyses revealed strong intercorrelations among several metrics—such as fixation duration, visit duration, and gaze frequency—that could weaken model robustness. By transforming these correlated variables into uncorrelated principal components, PCA clarified the data structure and enhanced analytical stability. For interpretability, Varimax rotation with Kaiser normalization was employed [54]. Varimax, an orthogonal rotation method, maximizes the variance of squared loadings within each component, simplifying the structure and highlighting the most influential variables [54]. This approach is particularly well suited for biometric datasets such as eye-tracking, where multiple indicators often reflect overlapping underlying behaviors [55]. Varimax rotation was chosen over oblique methods (e.g., Promax) because the aim was to extract distinct, non-overlapping dimensions of visual behavior—such as visual engagement or search effort—rather than correlated latent factors [56]. Kaiser normalization was included to prevent variables with larger variances from dominating the solution and to stabilize the variance structure during rotation [54]. Only components with eigenvalues greater than 1 were retained (Kaiser criterion), ensuring that each explained a meaningful share of the total variance. This approach produced a reduced set of orthogonal components representing the core dimensions of visual behavior in the packaging workflow, enabling the identification of visually demanding tasks and fatigue-prone conditions with greater precision.

#### 2.3.3. Visual Fatigue Trends and Critical Risk Detection

In the final phase of analysis, we examined how visual fatigue varied across operational and individual-level factors to identify the workflow conditions that posed the highest risk. Specifically, variations were analyzed in both objective eye-tracking metrics and subjective scale scores according to: moment within the shift (beginning vs. end), shift type (morning, afternoon, or night), day of the week (Monday–Friday), production line (Plant 4 Line 2, Plant 4 Line 3, Plant 4 Line 4, and Plant 8), batch size (1, 10, 100, or trays), operator age, and years of experience.

Prior to statistical comparisons, data normality was assessed using the Kolmogorov–Smirnov (KS) and Shapiro–Wilk (SW) tests [57]. For repeated-measures data, sphericity was evaluated using Mauchly’s Test of Sphericity when normality assumptions were satisfied [58]. Statistical significance testing was then applied to determine whether differences across groups or time points were meaningful. Correlation analyses were conducted according to data distribution: Pearson’s correlation coefficient was used for normally distributed variables, while Spearman’s rank correlation was applied when distributions deviated from normality [59].

This multi-dimensional analysis enabled the detection of fatigue progression patterns and the identification of conditions consistently associated with higher fatigue levels. By combining biometric trends with self-reported fatigue, the approach provided a robust framework for recognizing high-risk scenarios. These findings informed the ergonomic recommendations proposed to the industrial partner, ensuring that interventions targeted the areas with the greatest potential to reduce visual fatigue and improve operator comfort and performance.

## 3. Results

This section presents the main findings of the study, organized according to the three analytical domains described previously: (1) lighting level assessment, (2) eye-tracking metrics and Principal Component Analysis (PCA), and (3) visual fatigue trends and risk detection across operational and individual factors.

### 3.1. Lighting Levels

Figure 2 shows the lighting levels recorded across production lines throughout the workday. Each data point represents a measurement taken at the beginning of a trial, with colors and markers distinguishing Line 2 (blue squares), Line 3 (orange circles), Line 4 (green Xs), and Line 8 (purple triangles). The horizontal dashed lines denote the ISO/CIE 8995-1:2025 recommended illumination range for precision tasks: a minimum of 500 lux (blue) and a maximum of 1500 lux (red).

The results indicate that most measurements fell below the 500 lux threshold, with Line 8 consistently showing the lowest values, suggesting inadequate illumination for visually demanding inspection tasks. In contrast, Line 8 also exhibited occasional values exceeding 1500 lux, highlighting inconsistencies in local lighting conditions.

### 3.2. Eye-Tracker Results

We first assessed whether average pupil diameter could be reliably included in the analysis, given its sensitivity to external lighting conditions [60]. Normality was tested using the Kolmogorov–Smirnov (KS) and Shapiro–Wilk (SW) tests. Results yielded *p*-values of 0.070 (KS) and 0.012 (SW), indicating deviation from normality according to the Shapiro–Wilk test, which is more sensitive with smaller samples. Similarly, light intensity (lux) was non-normally distributed (*p* < 0.001 in both tests). Due to these deviations, Spearman’s rank correlation was applied to examine the relationship between pupil diameter and illumination. A significant negative correlation was observed (ρ = −0.506, *p* < 0.001), showing that higher light levels were consistently associated with smaller pupil sizes. This result is consistent with known physiological mechanisms, whereby pupil constriction regulates retinal exposure in response to increased ambient light [60]. Accordingly, pupil diameter was excluded from the final fatigue analysis, as its variation primarily reflected lighting conditions rather than operator fatigue, and no further correlations with environmental data were pursued.

The remaining seven eye-tracking metrics—Total Fixation Duration (Tot Fix Dur), Fixation Count, Total Visit Duration (Tot Visit Dur), Average Visit Duration (Avr Visit Dur), Visit Count, Percentage of Total Activity Time in the AoI, and Percentage of Gazes as Fixations—were analyzed using Principal Component Analysis (PCA). This approach improved interpretability by reducing multicollinearity and revealing latent dimensions of visual engagement. Dataset suitability was evaluated using the Kaiser–Meyer–Olkin (KMO) Measure of Sampling Adequacy and Bartlett’s Test of Sphericity. The KMO value was 0.538, slightly above the accepted minimum of 0.5, indicating marginal adequacy. Bartlett’s test was statistically significant (*p* < 0.001), confirming sufficient correlations among variables for PCA.

PCA revealed a clear structure in the dataset, identifying three principal components that together explained 93.64% of the total variance. Following the Kaiser criterion, only components with eigenvalues greater than 1 were retained. As shown in the scree plot (Figure 3), the first three components met this criterion, accounting for 52.38%, 27.46%, and 13.79% of the variance, respectively. The scree plot also displayed a distinct “elbow” at the third component, reinforcing this selection by illustrating the diminishing returns in variance explained beyond this point.

Varimax rotation with Kaiser normalization was applied, and only factor loadings above 0.60 are reported to highlight the most significant contributions of variables to each component. The rotated component matrix (Table 1) revealed three distinct components associated with eye-tracking parameters, which can be interpreted as follows:
Principal Component 1, “Overall Visual Engagement”: This component groups Total Visit Duration, Total Fixation Duration, Fixation Count, and Percentage of Total Activity Time in the AoI. Collectively, these metrics reflect the amount of visual attention and time devoted to task-relevant areas. High loadings on this component suggest sustained visual involvement, likely associated with task complexity or attentional demand. Therefore, this component is interpreted as a general indicator of visual engagement during operational activities.Principal Component 2, “Fixation Characteristics”: This factor includes Average Fixation Duration and Percentage of Gazes as Fixations, both of which describe the nature and stability of gaze behavior.Principal Component 3, “Visit Duration Pattern”: Driven solely by Average Visit Duration, this component highlights how long participants remained within each Area of Interest per visit.

Contextual interpretation is essential across all three Principal Components, as each captures a distinct dimension of visual behavior. Collectively, they provide valuable insights into shifts in visual strategy, attentional load, and early indicators of fatigue. Because this study focused specifically on visual fatigue, the interpretation of these components was grounded in prior literature on eye-movement behavior under fatigue conditions [61,62,63,64]. Table 2 summarizes the expected changes in key eye-tracking metrics as visual fatigue progresses.

As shown in Table 2, prior literature consistently reports that visual fatigue is characterized by increases in total and average fixation duration and blinking frequency, accompanied by decreases in fixation frequency and saccadic parameters. Principal Component 2 (Fixation Characteristics)—comprising Average Fixation Duration and Percentage of Gazes as Fixations—closely aligns with these empirically observed patterns. Both metrics capture elements of gaze stability and attentional persistence, which intensify as visual fatigue progresses. This convergence between the empirical loadings of PC2 and established theoretical expectations reinforces its interpretation as a valid and sensitive indicator of visual fatigue. Higher scores on this component reflect longer and more stable fixations, a behavior often associated with reduced efficiency in attentional modulation under fatigue.

Because the cognitive demands of the inspection task remain constant throughout the shift, systematic increases in PC2 values can be attributed to the accumulation of visual fatigue rather than variations in task complexity. Accordingly, PC2 was selected as the primary outcome variable in subsequent analyses examining how visual fatigue evolves under different contextual and operational conditions.

While all three components provide useful insights into eye-movement behavior, Principal Component 1—associated with overall visual engagement—appears to capture attentional demand or task structure rather than fatigue-specific effects. Principal Component 3, dominated by Average Visit Duration, may instead reflect individual exploration style or navigation strategy. Since neither PC1 nor PC3 aligns consistently with established fatigue markers in the literature, the analysis focused exclusively on PC2, which directly reflects fixation stability and duration—two parameters most reliably linked to the onset and progression of visual fatigue.

### 3.3. Visual Fatigue Trends

This section presents key findings on how visual fatigue, based on Principal Component 2 (Fixation Characteristics), varied according to shift timing, weekday, production line, batch type, age, and experience—highlighting critical risk scenarios for ergonomic intervention. Subjective data from the Visual Fatigue Scale were also incorporated to strengthen the analysis.

Normality of PC2 was assessed using both the Kolmogorov–Smirnov (KS) and Shapiro–Wilk (SW) tests. The results confirmed a normal distribution (KS: *p* = 0.200; SW: *p* = 0.122).

Following this validation, PC2 was analyzed across contextual and individual factors, including shift timing, production line, batch type, operator age, and experience. Using parametric tests and correlation analyses, no significant patterns emerged in relation to these variables.

A clear and consistent trend was, however, observed across the workweek. Because weekdays represent ordinal variables, Spearman’s rank correlation was applied, revealing a strong positive association between PC2 and the progression of the week (ρ = 0.529, *p* < 0.001). This finding indicates that visual fatigue accumulates over time, likely because of continuous exposure to visually demanding tasks without sufficient recovery.

Figure 4 illustrates this trend. At the beginning of the week (Monday), PC2 scores were negative, reflecting shorter and less stable fixations characteristic of a relatively rested visual state. As the week progressed, PC2 values steadily increased and shifted into positive territory, indicating longer and more stable fixations—consistent with the established physiological and behavioral signatures of accumulated visual fatigue.

Pairwise comparisons with Bonferroni correction revealed a statistically significant difference between Monday and Friday (*p* = 0.022), further supporting the hypothesis of fatigue accumulation.

An additional pattern emerged when analyzing the weekly trend of Overall Engagement (PC1) in relation to Fixation Characteristics (PC2), the designated indicator of visual fatigue. As illustrated in Figure 5, the mean plots suggest an inverse relationship between these two components.

This inverse trend reinforces the conceptual distinction between the components: while PC1 captures sustained attentional involvement, PC2 reflects fatigue-induced changes in gaze stability and fixation behavior. Their contrasting trajectories across the week empirically validate the use of PC2 as an indicator of visual fatigue and demonstrate how fatigue may undermine engagement during prolonged repetitive tasks.

The Visual Fatigue Scale provided complementary insights into operators’ subjective perceptions of visual strain, further supporting the biometric findings. Using Spearman’s rank correlation—appropriate for ordinal Likert-scale responses—significant associations were observed between perceived visual fatigue and both shift timing (ρ = 0.214, *p* = 0.048) and the specific point within the shift (ρ = 0.392, *p* < 0.001). These results indicate that perceived visual fatigue increases during later shifts (particularly the night shift) and accumulates as the workday progresses.

Another noteworthy finding from the Visual Fatigue Scale relates to the physical load associated with different lot sizes. In this context, Lot 1 refers to packaging single units, Lot 10 and Lot 100 correspond to batches of 10 and 100 units, and Tray 1 and Tray 2 involve handling larger trays with greater physical demands. As illustrated in Figure 6, the highest mean fatigue score was recorded for Tray 2. This batch type corresponded to operations in Production Line 8, where the lowest illumination levels had previously been observed. These results suggest a compounded effect of high physical workload and inadequate lighting on the accumulation of visual fatigue.

In addition, the Visual Fatigue Scale revealed a clear weekly trend, with perceived fatigue increasing from Monday (7.83) to Wednesday (7.90), followed by a drop on Thursday (6.50), and then rising again on Friday (6.92). This pattern closely mirrors the behavior of PC2 (Fixation Characteristics), reinforcing its validity as an indicator of visual fatigue. The decrease observed on Thursday can be explained by the fact that data collected on that day corresponded primarily to Lot 1 tasks, which are less visually demanding than the other operations (see Figure 6).

## 4. Discussion

This study highlights several critical risk factors contributing to visual fatigue in pharmaceutical inspection and packaging environments. Among the most pressing concerns was the suboptimal lighting observed in Production Line 8, where illumination levels frequently fell below the ISO/CIE 8995-1:2025 thresholds recommended for tasks requiring high visual precision. Inadequate lighting in such settings significantly elevates the risk of visual fatigue, potentially compromising both operator well-being and the accuracy essential for pharmaceutical inspections. Although some measurements temporarily exceeded the upper recommended limit, these instances were appropriately managed and did not directly affect operator visual comfort. As shown in Figure 6, perceived visual fatigue in Tray 1—corresponding to the highest recorded lux levels—remained relatively low compared to Tray 2 in Plant 8. This indicates that lighting variations were managed effectively, without compromising comfort.

To better understand how visual fatigue evolves under different work conditions, wearable eye-tracking technology (Tobii Pro Glasses 3) was combined with PCA, in line with Industry 5.0’s emphasis on real-time, worker-adaptive monitoring systems. Principal Component 2 (“Fixation Characteristics”)—comprising Average Fixation Duration and the Percentage of Gazes as Fixations—emerged as a reliable and objective indicator of visual fatigue. This component aligns closely with established physiological signatures of fatigue, such as longer and more stable fixations. Its validity was further reinforced by its empirical behavior: as Fixation Characteristics increased across the workweek, Principal Component 1 (“Overall Engagement”) simultaneously declined, reflecting the inverse relationship between attentional engagement and fatigue accumulation.

Subjective responses from the Visual Fatigue Scale further validated these biometric findings, together revealing four consistent trends:Visual fatigue increased steadily across the workweek, as indicated by rising PC2 scores and a concurrent decline in PC1.Perceived fatigue accumulated progressively within each shift, becoming more pronounced as operators advanced through their tasks—suggesting insufficient recovery during work hours.Night shifts were consistently associated with higher levels of perceived visual fatigue.Tray 2 tasks on Line 8 represented a critical risk scenario, combining high physical load with the lowest recorded lighting levels. Operators reported the highest fatigue scores in this condition, suggesting a compounded effect of physical workload and inadequate illumination.

These insights illustrate how Industry 5.0 tools, such as wearable sensors, can help identify high-risk conditions and inform proactive ergonomic strategies. Based on the findings, three targeted interventions are proposed, aligned with the human-centric design principles of Industry 5.0. First, enhancing local task lighting offers a cost-effective and high-impact solution: installing high-power desk lamps at workstations—particularly in underlit areas such as Line 8—can substantially reduce visual strain during detailed inspection tasks. Second, implementing adjustable overhead LED lighting systems can provide consistent, customizable illumination across shifts and task types, improving both visual comfort and long-term energy efficiency [65]. Finally, introducing structured visual breaks—particularly during night shifts or the latter half of work periods—can help mitigate fatigue accumulation. Evidence suggests that even short pauses every 20–30 min, allowing workers to rest their eyes by focusing on distant objects, reduce digital eye strain and support sustained visual performance [66].

It is important to note that these results are specific to the pharmaceutical packaging environment studied, where repetitive inspection and packaging tasks under variable lighting conditions strongly influenced the emergence of visual fatigue. While this sector provides a highly relevant context due to its demanding visual requirements, the findings should not be generalized to all industrial environments without caution. Instead, they highlight the potential of the proposed methodology, which can be adapted and validated in other operational settings through future research.

Finally, the relationship between visual fatigue, task engagement, and cognitive stress warrants discussion. In this study, the task itself remained constant across participants, consisting of repetitive inspection and packaging of pharmaceutical products. This ensured that task complexity and cognitive demands did not vary, minimizing the likelihood of cognitive stress as a confounding factor. Although physical demands differed slightly across batch types (1, 10, 100 units, or trays), the underlying activity was the same, reinforcing that observed variations were linked primarily to visual fatigue rather than task load. Moreover, the PCA helped separate Fixation Characteristics (PC2), which aligned with visual fatigue markers, from Overall Engagement (PC1), which reflected attentional involvement. Their inverse relationship across the week—validated by subjective fatigue reports—further supports the interpretation that PC2 captured fatigue-specific effects, while PC1 reflected broader engagement.

## 5. Conclusions

This study introduced a human-centered, sensor-based approach to assessing visual fatigue in pharmaceutical inspection and packaging environments—a domain where sustained attention, precision, and ergonomic sustainability are critical. By integrating wearable eye-tracking technology with subjective self-assessments, a novel, data-driven framework was developed to monitor fatigue in real time and identify high-risk working conditions. A key outcome was the identification of “Fixation Characteristics” as a robust indicator of visual fatigue, grounded both in theoretical literature and validated empirically through its inverse relationship with overall attentional engagement and alignment with self-reported fatigue levels. The convergence between sensor-based measures and subjective experience exemplifies the type of human-centric integration envisioned in Industry 5.0. Consistent with this paradigm, the findings support proactive ergonomic interventions that extend beyond productivity, emphasizing worker safety, comfort, and long-term well-being.

The study was limited to a single production facility within the pharmaceutical sector, which may constrain the generalizability of the results. The participant group was also strongly skewed toward female workers, reflecting the actual demographic composition of the packaging workforce. While this enhances ecological validity, it may also introduce gender-related bias, as physiological responses to fatigue could differ across populations. To address this, future studies should include more balanced participant groups to validate the findings across mixed populations and different workplace contexts. Encouragingly, preliminary data from a second case study in the automotive sector, conducted under real work conditions with an all-male participant group, revealed comparable fatigue patterns. These findings further support the cross-domain validity of the “Fixation Characteristics” component as a reliable indicator of visual fatigue, suggesting that the PCA-based methodology developed here holds promise across diverse industrial environments. Future research will aim to incorporate further factors known to influence visual fatigue, including glare, contrast ratios, circadian rhythm disruptions, workstation design, and cognitive workload. In addition, future work will focus on developing a quantitative predictive model of visual fatigue, combining eye-tracking features with the Visual Fatigue Scale to classify fatigue into different levels, similar to previous work on physical fatigue modeling [67,68].

Explicitly testing and validating this approach in different industrial settings beyond pharmaceutical packaging will be essential to establish broader applicability and to formulate standardized assessment criteria for visual fatigue in Industry 5.0 contexts. Nonetheless, we acknowledge that in more complex industrial settings, cognitive stress and task engagement may interact more strongly with visual fatigue. Future research should therefore integrate multimodal measures such as heart rate variability and EEG factors to better capture this interplay.

In conclusion, this study represents an important step toward bridging the gap between controlled laboratory research and the complexities of industrial reality. It provides empirical evidence that smart, non-intrusive technologies can be employed not only to monitor visual fatigue but also to guide proactive ergonomic strategies. In doing so, it advances the objectives of Industry 5.0: creating workplaces that are more efficient, adaptive, inclusive, and centered on human health and performance.

## Figures and Tables

**Figure 1 sensors-25-05702-f001:**
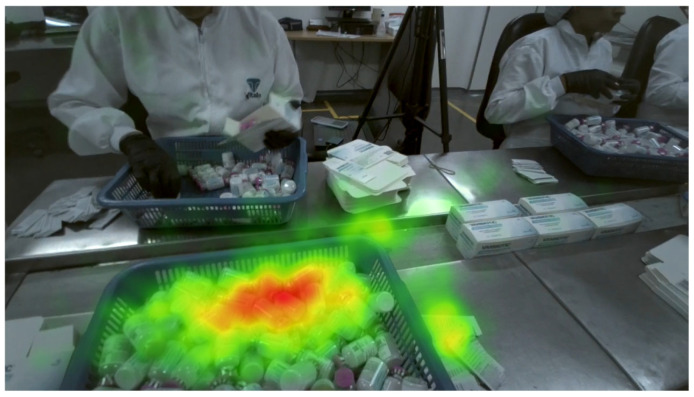
Representative heatmap from one participant’s eye-tracking data during pharmaceutical packaging tasks. Areas in red indicate regions of highest visual attention, while green areas reflect lower levels of focus.

**Figure 2 sensors-25-05702-f002:**
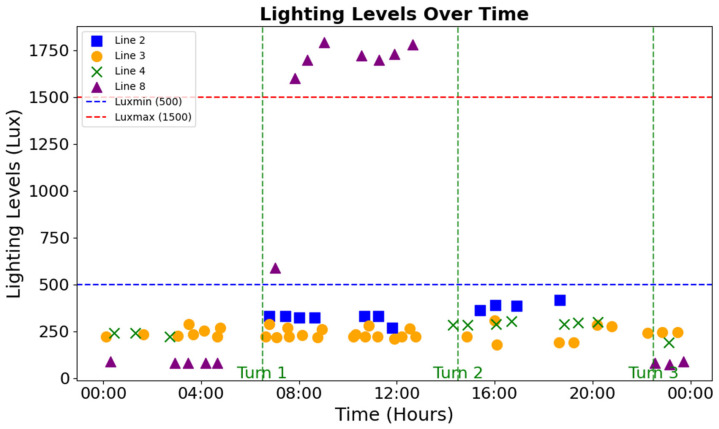
Lighting levels (in Lux) recorded at the beginning of each trial.

**Figure 3 sensors-25-05702-f003:**
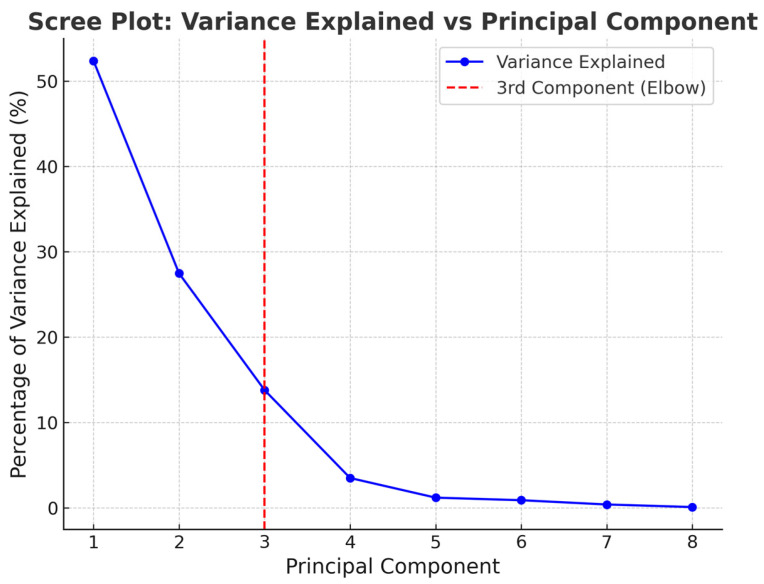
Scree plot of PCA for the seven eye-tracking metrics.

**Figure 4 sensors-25-05702-f004:**
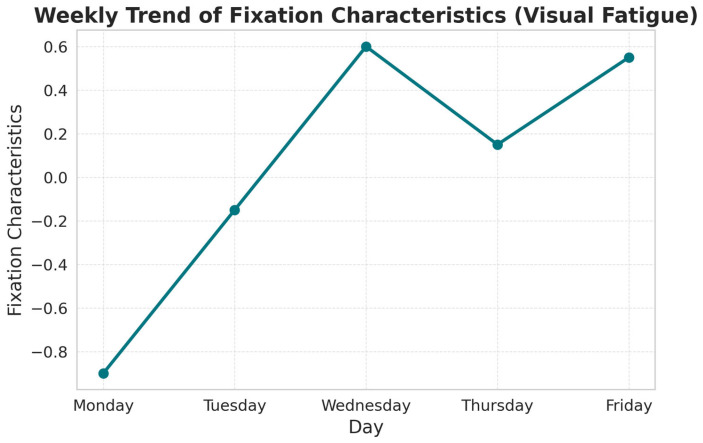
Weekly trend of mean Fixation Characteristics (PC2), the objective indicator of visual fatigue. Values begin negative on Monday, reflecting shorter and less stable fixations typical of a rested visual system, and increase steadily through Friday, indicating longer and more stable fixations characteristic of cumulative visual fatigue.

**Figure 5 sensors-25-05702-f005:**
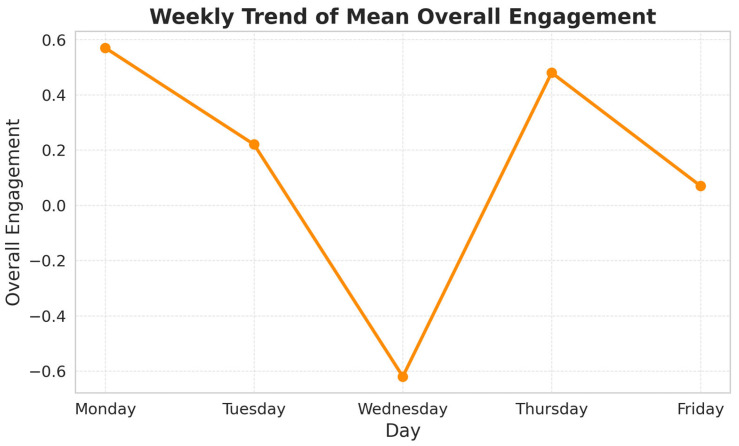
Weekly trend of mean Overall Visual Engagement (PC1). Engagement is highest at the beginning of the week, drops markedly by midweek (Wednesday), shows partial recovery thereafter, and tapers off again by Friday.

**Figure 6 sensors-25-05702-f006:**
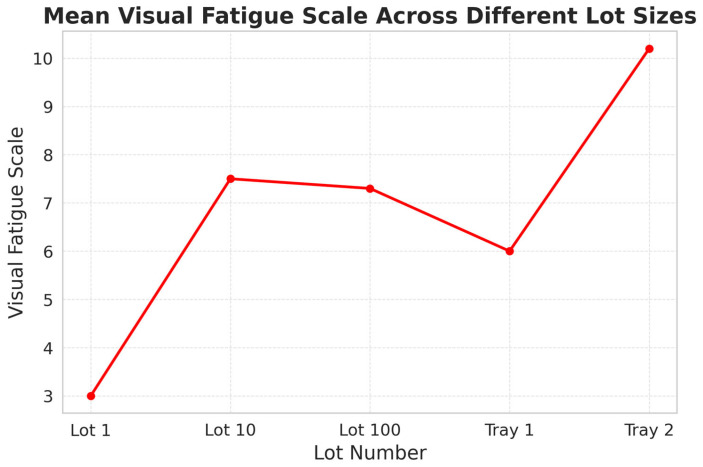
Mean Visual Fatigue Scale scores across different lot sizes. Perceived fatigue increased sharply from Lot 1 to Lot 10, stabilized through Lot 100, decreased slightly for Tray 1, and peaked sharply in Tray 2.

**Table 1 sensors-25-05702-t001:** Rotated Component Matrix for the seven eye-tracking metrics.

Variable		Principal Component	
	1	2	3
Tot Visit dur	0.981		
Tot fixation dur	0.969		
Fixation Count	0.944		
% total activity time	0.936		
% gazes are fixation		0.963	
Avr Fixation dur		0.904	
Avr Visit dur			0.942

**Table 2 sensors-25-05702-t002:** Eye movement parameters and their expected change with visual fatigue.

Eye Movement Parameters	Change with Visual Fatigue
Fixation frequency	↓
Total fixation duration	↑
Average fixation duration	↑
Percentage of Gazes as Fixations	↑
Average fixation angle: left eye	~
Average fixation angle: right eye	~
Total saccade duration	↓
Average saccade duration	↓
Average saccade magnitude: left eye	↓
Average saccade magnitude: right eye	↓
Maximum saccade magnitude: left eye	↓
Maximum saccade magnitude: right eye	↓
Blinking frequency	↑
Average blinking duration	↑
Average rotation angle: left eye	~
Average rotation angle: right eye	~
Pupil diameter	↓

↑: Increase with visual fatigue; ↓: Decrease with visual fatigue; ~: Little to no change.

## Data Availability

The anonymized data files are available in the public Zenodo repository: https://doi.org/10.5281/zenodo.15705327.

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
