# Peer review of "Monitoring Visual Fatigue with Eye Tracking in a Pharmaceutical Packing Area"

_sensors, 2025, doi:10.3390/s25185702_

Round 1
Reviewer 1 Report
Comments and Suggestions for Authors
The study evaluates visual fatigue in a pharmaceutical packaging workplace using a combination of eye tracking with Tobii Pro Glasses 3, self-report questionnaires, and lux-level measurements. Although the sample size of 43 participants is relatively large, 42 of them are female, indicating a gender imbalance. Nevertheless, I think the findings are valuable and worth sharing. However, the following points require revision:
- Comparison with existing visual fatigue indicators is needed.
There are several established measures of visual fatigue, such as critical flicker fusion frequency and accommodative function tests. The authors should clearly state in what ways their proposed method is effective in comparison to these conventional approaches. - Clarification of subjective data collection.
While it is useful if visual fatigue can be evaluated from eye-tracking metrics, it remains unclear how frequently and in what manner subjective data were collected for real-time assessment of visual fatigue. - Clarification of experimental setup.
The authors should clarify the experimental setup. Including diagrams or illustrations of the factory environment studied would also be helpful. - Consideration of participant bias.
The participant group is strongly skewed toward female workers. This dataset is valuable and reflects the reality of that workplace. However, since the study deals with physiological data such as visual fatigue, the authors should acknowledge this limitation explicitly in their conclusions and avoid overgeneralization. - Scope limitation of the study.
Visual fatigue arises in many contexts, but the outcomes of this study are highly specific to the industrial packaging environment. In the highlights, contributions, and summary of results, the authors should explicitly state that the findings are limited to this specific setting. - Relationship with cognitive stress and task engagement.
In the paper, the relationship between cognitive stress, task engagement, and the reported results remains unclear. While the separation of “visual engagement” and “visual fatigue” through statistical methods can be interpreted as an attempt to avoid conflating visual fatigue with cognitive elements, the authors should explicitly address how they avoided such confounding. Furthermore, it would strengthen the paper if they discuss how the current findings can be interpreted in light of potential overlaps between visual fatigue, task focus, and cognitive demands.
Author Response
Comment 1: Figures and tables can be improved.
Response 1: We thank the reviewer for this valuable observation. All tables and figures have been carefully revised to improve their clarity, readability, and consistency.
Comment 2: Comparison with existing visual fatigue indicators is needed.
There are several established measures of visual fatigue, such as critical flicker fusion frequency and accommodative function tests. The authors should clearly state in what ways their proposed method is effective in comparison to these conventional approaches.
Response 2: We thank the reviewer for this important suggestion. We agree that comparing our approach with conventional visual fatigue measures helps to clarify its contribution. Established methods, such as critical flicker fusion frequency (CFF) and accommodative function tests, have been widely used in controlled laboratory or clinical settings. However, these methods are difficult to apply in real industrial environments because they require specialized equipment, interrupt normal workflows, and are not well suited for continuous or non-intrusive monitoring.
In contrast, our proposed methodology—based on eye-tracking features (Fixation Characteristics, PC2) combined with subjective self-reports—provides a practical and scalable solution for shop-floor applications. It enables continuous, real-time data collection without disrupting ongoing operations, and it directly links visual behavior to the actual work context. This integration of objective biometric signals with contextualized subjective measures aligns with Industry 5.0 principles by supporting human-centered, adaptive monitoring. We have now emphasized this comparison and its practical implications in the revised manuscript (line 129, Introduction).
Comment 3: Clarification of subjective data collection.
While it is useful if visual fatigue can be evaluated from eye-tracking metrics, it remains unclear how frequently and in what manner subjective data were collected for real-time assessment of visual fatigue.
Response 3: We thank the reviewer for pointing out the need for clarification. Each participant completed the questionnaire once before the start of each eye-tracking session. Since two trials were conducted per participant (at the beginning and at the end of the shift), this resulted in two subjective assessments per participant, providing complementary data to the eye-tracking measurements. This clarification has been added in the revised manuscript (line 291, Data collection).
Comment 4: Clarification of experimental setup.
The authors should clarify the experimental setup. Including diagrams or illustrations of the factory environment studied would also be helpful.
Response 4: We thank the reviewer for this helpful suggestion. To improve clarity, we have added Figure 1, which presents a representative heatmap generated from one participant’s eye-tracking data. The illustration shows the operator’s visual attention focused on the trays of pharmaceutical products prior to packaging, under real shop-floor conditions. This figure helps to contextualize the experimental setup, highlighting the actual workstation layout, the materials being handled, and the visual demands of the task. A short explanation related to Figure 1 has also been added in the revised manuscript (line 211, Experimental Setup).
Comment 5: Consideration of participant bias.
The participant group is strongly skewed toward female workers. This dataset is valuable and reflects the reality of that workplace. However, since the study deals with physiological data such as visual fatigue, the authors should acknowledge this limitation explicitly in their conclusions and avoid overgeneralization.
Response 5: We thank the reviewer for this important observation. Indeed, the participant group in this study was strongly skewed toward female workers, which reflects the actual demographic composition of the workforce in the packaging area studied. However, as the reviewer notes, this distribution may limit the generalizability of the findings, particularly since physiological responses to visual fatigue could vary across populations. We have now explicitly acknowledged this limitation in the Conclusions section (line 666) and emphasized that future work should include more balanced participant groups to validate the findings across mixed populations.
Comment 6: Scope limitation of the study.
Visual fatigue arises in many contexts, but the outcomes of this study are highly specific to the industrial packaging environment. In the highlights, contributions, and summary of results, the authors should explicitly state that the findings are limited to this specific setting.
Response 6: We thank the reviewer for this important observation. We agree that the outcomes of this study are highly specific to the industrial packaging environment. To address this, we have revised the manuscript to explicitly acknowledge this limitation in both the Discussion (line 630) and Conclusions (line 664). We also highlight that, while the results are limited to this particular setting, they open the door for future research in other industrial contexts to further validate and extend the methodology.
Comment 7: Relationship with cognitive stress and task engagement.
In the paper, the relationship between cognitive stress, task engagement, and the reported results remains unclear. While the separation of “visual engagement” and “visual fatigue” through statistical methods can be interpreted as an attempt to avoid conflating visual fatigue with cognitive elements, the authors should explicitly address how they avoided such confounding. Furthermore, it would strengthen the paper if they discuss how the current findings can be interpreted in light of potential overlaps between visual fatigue, task focus, and cognitive demands.
Response 7: We thank the reviewer for this insightful comment. We agree that distinguishing visual fatigue from cognitive stress and task engagement is essential to avoid confounding interpretations. In our study, this was addressed in several ways. First, the task itself remained constant throughout the study, as all participants performed the same repetitive packaging and inspection activity (placing drugs into boxes or trays). Since task complexity and cognitive demands did not vary, we consider cognitive stress to be minimal in this context and unlikely to confound the results. Although the physical demand could vary slightly depending on the batch type (1, 10, 100 units, or trays), the core activity remained the same—repetitive inspection and packaging—ensuring that variations in fatigue were not attributable to differences in cognitive workload.
Second, we employed Principal Component Analysis (PCA) to separate correlated eye-tracking variables into orthogonal dimensions. This enabled us to isolate Fixation Characteristics (PC2) as the component most closely aligned with established markers of visual fatigue (longer and more stable fixations). In contrast, Overall Engagement (PC1) captured general attentional involvement with the task, which we interpret as reflecting task focus rather than fatigue. This statistical separation reduces the risk of conflating visual fatigue with engagement or stress.
Third, we triangulated objective data with subjective reports. While PC2 increased across the week (consistent with cumulative fatigue), PC1 decreased, reflecting reduced engagement—an inverse relationship that reinforces the interpretation of PC2 as fatigue-related rather than engagement-related. Subjective fatigue reports from the Visual Fatigue Scale also aligned with PC2 but not with PC1, further validating this distinction.
In the revised manuscript, we have expanded the Discussion (line 637) to explicitly state that, given the repetitive and uniform nature of the tasks, cognitive stress was not considered a major factor in this study. We also highlight that while fatigue and engagement are distinct constructs, they interact dynamically: prolonged fatigue can undermine engagement, while high task focus may temporarily mask early signs of fatigue. As noted in the Conclusions (line 682), future research should incorporate additional multimodal measures (e.g., heart rate variability, EEG-based workload indices) to more fully explore the interplay between visual fatigue, task engagement, and cognitive stress in more complex industrial settings.
Reviewer 2 Report
Comments and Suggestions for Authors
The paper presents a method for assessing visual fatigue by combining self-report questionnaires, eye-tracking data, and environmental lighting measurements. Research results show that smart, non-intrusive technologies can achieve real-time fatigue monitoring and early warning. This study is clearly valuable for enhancing operator well-being and inspection performance. I have a couple of questions as described below.
1. In Fig. 3, why did the fixation characteristics decrease on Thursday? Similarly, why did overall engagement rise sharply on Thursday in Fig. 4?
2. This paper reveals the trend relationship between visual fatigue and eye-tracking data. Is it possible to establish a quantitative model between the two and formulate an assessing standard?
Author Response
Comment 1: The English could be improved to more clearly express the researc
Response 1: We thank the reviewer for this observation. The manuscript has been carefully revised throughout to improve the clarity of expression and readability.
Comment 2: In Fig. 3, why did the fixation characteristics decrease on Thursday? Similarly, why did overall engagement rise sharply on Thursday in Fig. 4?
Response 2: Thank you very much for highlighting this point, as addressing it further strengthens the interpretation of PC2 (Fixation Characteristics) as the most reliable indicator of visual fatigue. The decrease observed on Thursday can be explained by the fact that the data collected on that day corresponded primarily to Lot 1 tasks, which are less visually demanding than the other operations (see Fig. 6). This interpretation is also supported by the subjective Visual Fatigue Scale results, where the average perceived fatigue decreased from Wednesday (7.90) to Thursday (6.50), before increasing again on Friday (6.92).
The sharp rise in Overall Engagement (PC1) observed on Thursday in Fig. 4 reflects this inverse relationship: as visual fatigue (PC2) decreases under less demanding conditions, attentional engagement (PC1) temporarily recovers. This inverse behavior between PC1 and PC2 further validates our decision to use PC2 as the main proxy for visual fatigue, since PC1 captures general attentional involvement rather than fatigue-specific effects. We have also included in line 573 a short explanation of this point in the revised manuscript.
Comment 3: This paper reveals the trend relationship between visual fatigue and eye-tracking data. Is it possible to establish a quantitative model between the two and formulate an assessing standard?
Response 3: Thank you very much for this insightful suggestion. Indeed, the development of a quantitative model to establish standardized levels of visual fatigue would represent a valuable advancement. In previous work, we have developed a classification model for physical fatigue using smartwatch data, categorizing fatigue into different levels (very high, high, medium, low, very low). A similar approach could be pursued here by combining eye-tracking features with the Visual Fatigue Scale for data labeling and applying supervised machine learning classification algorithms. We have now included this perspective as a future research direction in the revised manuscript (line 680, Conclusions), together with references to our previous work on physical fatigue classification.
Round 2
Reviewer 1 Report
Comments and Suggestions for Authors
I have confirmed that the authors have appropriately revised the manuscript in response to the conditions I suggested; therefore, I recommend the paper for acceptance.